# Divergent Metabolomic Signatures of TGFβ2 and TNFα in the Induction of Retinal Epithelial-Mesenchymal Transition

**DOI:** 10.3390/metabo13020213

**Published:** 2023-01-31

**Authors:** Pei Qin Ng, Magali Saint-Geniez, Leo A. Kim, Daisy Y. Shu

**Affiliations:** 1Department of Plant Science, University of Cambridge, Downing Street, Cambridge CB2 3EA, Cambridgeshire, UK; 2Schepens Eye Research Institute of Mass Eye and Ear, Boston, MA 02114, USA; 3Department of Ophthalmology, Harvard Medical School, Boston, MA 02114, USA; 4School of Biological Sciences, The University of Adelaide, Adelaide, SA 5005, Australia; 5South Australian Health and Medical Research Institute (SAHMRI), Adelaide, SA 5000, Australia

**Keywords:** retinal pigment epithelium (RPE), metabolomics, metabolism, mitochondria, tumor necrosis factor-alpha (TNFα), transforming growth factor-beta (TGFβ), epithelial-mesenchymal transition (EMT), OXPHOS, glycolysis, age-related macular degeneration, proliferative vitreoretinopathy

## Abstract

Epithelial-mesenchymal transition (EMT) is a dedifferentiation program in which polarized, differentiated epithelial cells lose their cell-cell adhesions and transform into matrix-producing mesenchymal cells. EMT of retinal pigment epithelial (RPE) cells plays a crucial role in many retinal diseases, including age-related macular degeneration, proliferative vitreoretinopathy, and diabetic retinopathy. This dynamic process requires complex metabolic reprogramming to accommodate the demands of this dramatic cellular transformation. Both transforming growth factor-beta 2 (TGFβ2) and tumor necrosis factor-alpha (TNFα) have the capacity to induce EMT in RPE cells; however, little is known about their impact on the RPE metabolome. Untargeted metabolomics using high-resolution mass spectrometry was performed to reveal the metabolomic signatures of cellular and secreted metabolites of primary human fetal RPE cells treated with either TGFβ2 or TNFα for 5 days. A total of 638 metabolites were detected in both samples; 188 were annotated as primary metabolites. Metabolomics profiling showed distinct metabolomic signatures associated with TGFβ2 and TNFα treatment. Enrichment pathway network analysis revealed alterations in the pentose phosphate pathway, galactose metabolism, nucleotide and pyrimidine metabolism, purine metabolism, and arginine and proline metabolism in TNFα-treated cells compared to untreated control cells, whereas TGFβ2 treatment induced perturbations in fatty acid biosynthesis metabolism, the linoleic acid pathway, and the Notch signaling pathway. These results provide a broad metabolic understanding of the bioenergetic rewiring processes governing TGFβ2- and TNFα-dependent induction of EMT. Elucidating the contributions of TGFβ2 and TNFα and their mechanistic differences in promoting EMT of RPE will enable the identification of novel biomarkers for diagnosis, management, and tailored drug development for retinal fibrotic diseases.

## 1. Introduction

Epithelial-mesenchymal transition (EMT) is a process in which polarized, differentiated epithelial cells dedifferentiate into matrix-producing mesenchymal cells [1]. This complex biological process is defined by a cascade of cellular and molecular events, including loss of cell–cell adhesions, excessive deposition of extracellular matrix (ECM) proteins, profound cytoskeletal reorganization, increased invasiveness and contractility, and reduced expression of epithelial markers such as E-cadherin and zonula occludens (ZO)-1 [2]. EMT of retinal pigment epithelial (RPE) cells plays an integral role in many retinal pathologies, including dry age-related macular degeneration (AMD) [3,4,5,6,7,8], wet AMD [9,10], proliferative vitreoretinopathy (PVR) [8,11,12,13,14], and diabetic retinopathy (DR) [15].

Metabolic reprogramming has emerged as a prominent hallmark of EMT, particularly in the field of cancer metastasis [16]. Since the seminal observations of the Warburg effect, which found that cancer cells preferentially use glycolysis over mitochondrial oxidative phosphorylation (OXPHOS) for energy generation despite the presence of oxygen [17], numerous examples of dysregulated metabolic pathways in EMT have been documented in glycine metabolism [18], glutamine metabolism [19], the pentose phosphate pathway [19], and lipid metabolism [20]. With significant advances in the field of metabolic reprogramming in EMT during cancer, efforts have been directed towards deciphering the metabolic alterations in retinal EMT. Work in our laboratory has explored the metabolic changes induced by two key EMT inducers: transforming growth factor-beta 2 (TGFβ2) and tumor necrosis factor-alpha (TNFα). While both cytokines potently induce EMT in RPE, they exhibit dramatically distinct bioenergetic profiles: TGFβ2 suppresses mitochondrial respiration and enhances glycolytic capacity [14], whereas TNFα enhances mitochondrial respiration and reduces glycolysis [8].

Both TGFβ2 and TNFα have been implicated in the pathogenesis of AMD. Persistent expression of the TGFβ pathway in RPE is associated with choroidal neovascularization and geographic atrophy [3], and a polymorphism in the TGFβ receptor type I (TGFBR1) gene is linked to an increased risk of developing AMD [21]. In contrast, TNFα drives the inflammatory component of AMD and mediates the formation of choroidal neovascular membranes by regulating the expression of vascular endothelial growth factor (VEGF) in RPE [22]. Patients with enhanced serum levels of pro-inflammatory cytokines such as interleukin-6 (IL-6) and TNFα respond more favorably to anti-VEGF therapy [23]. Single nucleotide polymorphisms in the TNFα gene have been observed in AMD patients [24].

Among the suite of omics technologies available to elucidate the molecular and biochemical perturbations underpinning retinal EMT, metabolomics is particularly powerful [25]. Metabolomics, defined as the comprehensive analysis of the multitude of native small molecules (metabolites) in a biological specimen, is a rapidly expanding area. Alterations in metabolic fluxes are directly linked to changes in the level of intermediates in affected metabolic pathways and are downstream of changes in gene expression as well as post-transcriptional and post-translational events [16].

In this study, we harnessed the broad scope of untargeted metabolomics to characterize the metabolic reprogramming associated with TNFα- and TGFβ2-induced EMT of RPE. The complete metabolome comprises the endo-metabolome (metabolites within the cell) and the exo-metabolome (metabolites in the extracellular medium) [26]. The untargeted approach enables the generation of an unbiased and comprehensive understanding of how the endo- and exo-metabolome are affected in RPE mesenchymal transition with two distinct stimulators of EMT. In this study, we evaluated both the cellular and media metabolites of H-RPE treated with TNFα or TGFβ2 compared to untreated control H-RPE. We profiled 638 metabolites in both the cellular and secreted metabolome associated with TNFα and TGFβ2 treatments of primary human fetal RPE cells. We aimed to determine the significantly altered metabolites during retinal EMT by comparing the altered metabolite profiles between TNFα-treated or TGFβ2-treated cells vs. untreated control cells as well as differences between the metabolite profiles of TNFα and TGFβ2 treatments.

## 2. Materials and Methods

### 2.1. Cell Culture

Primary human fetal RPE (H-RPE, Lonza, Walkersville, MD, USA) were cultured in an RtEGM Retinal Pigment Epithelial Cell Growth Medium supplemented with RtEGM SingleQuots (Lonza, Walkersville, MD, USA), as described previously [8,14]. H-RPE were plated in T25 flasks and maintained in a humidified incubator at 37 °C and 5% CO_2_. Cells were passaged 1:3 up to a maximum of 5 passages. Half of the media was changed every two–four days for a month to allow sufficient time for RPE maturation, including pigment accumulation. Cells were tested monthly for mycoplasma contamination (Mycoplasma PCR Test, Applied Biological Materials). Following maturation, cells were serum-starved for two days before treatment for five days with recombinant human TGFβ2 (Peprotech, Rocky Hill, NJ, USA) or TNFα (Peprotech, Rocky Hill, NJ, USA), both at 10 ng/mL in serum-free media. Growth factors were added freshly each day. This dosage and duration are required for proper mesenchymal transition of H-RPE, as shown in our previous studies [8,14]. Analysis was performed on three groups (control, TGFβ2 and TNFα) with N = 8 in each group.

### 2.2. Sample Preparation

Sample preparation for analysis of secreted metabolites involved collecting cell culture media (1.5 mL) from each flask and clarifying by centrifugation at 200× *g* for 30 s. One mL of supernatant was transferred into a fresh tube, snap-frozen in liquid nitrogen, and stored at −80 °C until analysis.

For sample preparation of cellular metabolites, H-RPE were rinsed in HEPES-buffered saline and then trypsinized using the ReagentPackTM Subculture Reagents (Lonza, Walkersville, MD, USA) for 5 min in a 37 °C incubator. Cells were then rinsed in trypsin neutralizing solution and centrifuged at 200× *g* for 3.5 min. The cell pellet was rinsed in phosphate buffered saline (PBS), and cells were counted using a hemocytometer. Five million cells per sample were centrifuged at 200× *g* for 3.5 min, and the final cell pellet was snap-frozen in liquid nitrogen and stored at −80 °C until analysis.

### 2.3. Metabolomics Data Acquisition Using Gas Chromatography Time-of-Flight Mass Spectrometry (GC-TOF MS)

Cellular and media samples were shipped on dry ice to the NIH West Coast Metabolomics Center (UC Davis, LA, USA) for sample processing and analysis as described in [27]. Samples were extracted using 1 mL of 3:3:2 acetonitrile:isopropanol:H_2_O (*v/v/v*). Half of the sample was completely dried and then derivatized to increase the volatility and stability for subsequent GC-TOF MS data acquisition. The derivatization process involved adding 10 uL of 40 mg/mL methoxyamine to pyridine and shaking at 30 °C for 1.5 h. Following this, 91 uL of a mixture of MSTFA and FAMEs was added to each sample with shaking at 37 °C for 0.5 h to complete derivatization. Samples were then vialed, capped, and injected onto the instrument. Samples were run on a 7890A GC coupled with a LECO TOF. The derivatized sample (0.5 uL) was injected using a splitless method onto a RESTEK RTX-5SIL MS column with an Integra-Guard at 275 °C with a helium flow of 1 mL/min. The GC oven was set to hold at 50 °C for 1 min and then ramp to 20 °C/min to 330 °C and then hold for 5 min. The transfer line was set to 280 °C, while the EI ion source was set to 250 °C. Metabolites were identified as peaks characterized by mass-over-charge ratio (*m/z*) and retention time. The parameters for mass spectrometry were set to collect data from 85 *m/z* to 500 *m/z* at an acquisition rate of 17 spectra/second. Raw data was deconvoluted using ChromaTOF software v2.32 (LECO Corporation) and processed by the BinBase algorithm for compound identification and quantification.

### 2.4. Metabolomics Data Analysis

Metabolomics data were first processed using an in-house R script. Processed data were first inspected for sample clustering and variance difference via Principal Component Analysis (PCA) and Partial Least Square Discriminant Analysis (PLSDA), in which PLSDA plots were generated using the R package mixOmics v6.18.1 [28]. Raw *p*-values were adjusted using the Benjamini–Hochberg correction and then tested for significance using the Welch method for metabolite profiling using the R package Omu v1.0.6 [29]. Differential metabolite analysis between the TNFα- and TGFβ2-treated samples compared to control (untreated) samples for both cell and media metabolites was generated by the R package Omu using the *count_fold_change* function, with the log FoldChange (FC) >1.5 or <−1.5 (TNFα-treated H-RPE vs. control H-RPE), logFC > 1.0, and logFC < −1.0 (TGFβ2-treated H-RPE vs. control H-RPE), with a Benjamini–Hochberg adjusted *p*-value cut off < 0.05 [29].

The Enriched Pathway Network Analysis was generated using the KEGG database of *Homo sapiens* [30], which was loaded locally using the functions *buildGraphFromKEGGREST()* and *buildDataFromGraph()* in the R package FELLA v1.14.0 [31]. The lists of metabolites that were significantly different from the TNFα-treated vs. control and TGFβ2-treated vs. control were extracted. The KEGG compound hierarchy was assigned to the extracted list of metabolites for the two comparisons using the function *defineCompounds()* in FELLA by mapping the metabolite compounds against the loaded database. The KEGG-assigned metabolomics data were then used as input for pathway enrichment analysis using the undirected heat diffusion model followed by statistical normalization using Z-scores for sub-network analysis in the R package FELLA v1.14.0 [31]. For the pathway enrichment analysis, metabolites with logFC > 1.5 or <−1.5 (TNFα-treated vs. control), logFC > 1.0, and logFC < −1.0 (TGFβ2-treated vs. control), with a Benjamini–Hochberg adjusted *p*-value < 0.05 were included. The purpose of two different fold-change cutoffs was due to differences in metabolite changes between TNFα-treated and TGFβ2-treated H-RPE, enabling the capture of sufficient differential metabolites in each group for generating a comprehensive network analysis. The enrichment analysis outputs were then mapped to the *Homo sapiens* (hsa) KEGG graphs and subsequently used for network analysis. Optimal visualization of the metabolic network graphs was generated with the number of nodes limit (nlimit) of 250 for TNFα-treated H-RPE vs. control and nlimit of 160 for TGFβ2-treated H-RPE vs. control using the *generateResultsGraph()* in FELLA. KEGG IDs unmapped to the KEGG graphs were retrieved and searched against the KEGG pathway database (https://www.genome.jp/kegg/pathway.html, accessed on 28 December 2022).

## 3. Results

### 3.1. TNFα- or TGFβ2-Treated H-RPE Exhibited Distinct Cellular Metabolomic Signatures

Principal Component Analysis (PCA) was performed for both cell and media samples to generate clustering patterns of metabolite profiles and detect outliers. For cellular metabolites, three clusters were identified in the PCA score plot (Figure 1A), indicating significant differences between control, TNFα, and TGFβ2. One sample in the TGFβ2 cellular metabolite group was identified as an outlier by cluster analysis and excluded in subsequent analyses. However, for the secreted metabolites, no definitive clusters were observed (Appendix A). We further confirmed the lack of separation between these two groups by performing PLSDA analysis, where overlap occurred between the clusters from media derived from untreated cells (control) and media derived from TNFα and TGFβ2. However, distinct clusters were observed between media samples derived from TNFα- and TGFβ2-treated cells (Appendix A). Analysis of the differential metabolites of media samples showed that for each treatment, only one primary metabolite was significantly different. For media derived from TNFα-treated cells, tartaric acid showed a significant increase compared to media derived from untreated H-RPE cells (logFC > 1.0, *p*-value < 0.05); in media samples derived from TGFβ2-treated cells, isobutylamine showed significant downregulation compared to control (logFC < −1.0, *p*-value < 0.05).

A total of 638 metabolites were detected in both cell metabolite samples; 188 were annotated as primary metabolites and displayed as a heatmap (Figure 1B). A comparison of cellular metabolite changes between TNFα-treated H-RPE and control cells revealed that peptides and organic acids were among the most prominent metabolite classes with significant changes (*p*adj < 0.05, *p*-value < 0.05) at ~ 30% and ~32% percent, respectively (Figure 1C). Other metabolite classes that showed significant variation following TNFα treatment were vitamins and cofactors (~26%), lipids (~9%), and nucleic acids (~5%). Comparison of cellular metabolite changes between TGFβ2 and control revealed that lipids and organic acids were most prominently affected at ~50% and ~42%, respectively (Figure 1C). Nucleic acids contributed only 9% of the metabolites altered in the TGFβ2 group.

Since organic acids, nucleic acids, and lipids were common metabolite classes affected in both TNFα and TGFβ2 groups, we examined the compounds in these classes that were significantly changed due to either TNFα- and TGFβ2-treatment (Appendix A). Arachidonic acid (lipid metabolite class) and adenosine (nucleic acid metabolite class) were upregulated in TNFα-treated H-RPE cells, while uridine (nucleic acid metabolite class) was downregulated in TNFα-treated H-RPE cells compared to control. All organic acid compounds (lactic acid, parabanic acid, and malonic acid) were higher in TNFα-treated H-RPE cells compared to control. In TGFβ2-treated H-RPE cells, lipids (linoleic acid and palmitoleic acid) were upregulated, whereas malic acid (organic acid metabolite class) and urea (nucleic acid metabolite class) were downregulated with TGFβ2 treatment compared to control cells.

For visualization of the statistical significance of differential metabolite alterations, volcano plots were generated by comparing the fold change size (*x*-axis) to the adjusted *p* value (*y*-axis) for both TNFα- and TGFβ2-treated cells compared to the control. The plot for TNFα-treated cells highlighted a significant upregulation of galactose 6-phosphate and nicotinamide along with a downregulation of putrescine, uridine, cadaverine, syringic acid, and trans−4−hydroxy−L−proline (Figure 2A). Alterations in these metabolites across each sample in the TNFα and control groups are depicted in the heatmap (Figure 2B). 

The volcano plot comparing TGFβ2 to control cells revealed a significant upregulation of linoleic acid and palmitoleic acid as well as a downregulation in malic acid, kynurenine, and urea (Figure 3A), as further depicted in the heatmap (Figure 3B). A comparison of metabolites between TNFα-and TGFβ2-treated cells highlights upregulation of ribose-5-phosphate, ribulose-5-phosphate, nicotinamide, and galactose-6-phosphate in TNFα-treated cells (Figure 4), whereas palmitoleic acid and linoleic acid were upregulated in TGFβ2-treated cells (Figure 4). Uridine, trans-4-hydroxy-L-proline, syringic acid, putrescine, and cadaverine were downregulated in TNFα-treated cells, whereas urea, malic acid, and kynurenine were significantly downregulated in TGFβ2-treated cells (Figure 4).

### 3.2. Metabolite Pathway Enrichment Analysis Revealed Distinct Regulatory Networks for TNFα and TGFβ2

Metabolite pathway network associations for both TNFα- and TGFβ2-treated cells vs. control were identified and visualized using pathway enrichment analysis. We first performed pathway enrichment followed by network analysis with metabolites by inputting the list of significantly different metabolites in the respective comparisons. In TNFα-treated H-RPE cells, 19 key pathways were enriched (Figure 5A) with 10 compounds. The pentose phosphate pathway (putrescine) and galactose metabolism pathway (D-galactose-6-phosphate), both part of carbohydrate metabolism, were shown to be enriched with TNFα treatment. Other primary enriched pathways included protein digestion and adsorption (cadaverine), the nuclear factor kappa B (NF-κB) signaling pathway (nicotinamide), Nod-like receptor signaling pathway, base excision repair (nicotinamide), nucleotide and pyrimidine metabolism (uridine), purine metabolism (parabanic acid), and arginine and proline metabolism (trans-4-hydroxy-L-proline). These primary enriched pathways impacted several secondary pathways, such as HIF-1 signaling, apoptosis, and hematopoietic cell lineage pathways (Figure 5A).

For TGFβ2-treated H-RPE cells, 18 key pathways were enriched in the network analysis with three compounds: urea, linoleic acid, and palmitoleic acid (Figure 5B). Primary pathways directly associated with the altered metabolites were the linoleic acid pathway (linoleic acid), the Notch signaling pathway (urea), and the fatty acid synthesis pathway (palmitoleic acid). These were then shown to impact a cascade of 15 secondary pathways, including the TGFβ2 signaling pathway, Wnt signaling pathway, Hippo signaling pathway, and ubiquitination pathway (Figure 5B).

We further investigated the three compounds that did not directly map onto the hsa KEGG graph within the FELLA package: syringic acid (C10833) for TNFα-treated H-RPE cells as well as kynurenine (C01718) and malate (C00711) for TGFβ2-treated H-RPE cells. These compounds were manually inputted into the KEGG pathway database to render the pathway network. Syringate, a benzoate ester of syringic acid, is part of the aminobenzoate pathway (Appendix A). (S)-malate is involved in glyoxylate and dicarboxylate metabolism, which feeds into the carbohydrate metabolism cycle (Appendix A). L-kynurenine, an isomer of kynurenine, is involved in the tryptophan signaling pathway (Appendix A).

## 4. Discussion

EMT is a metabolically demanding process and, as such, requires dramatic reprogramming of cellular metabolism. Here we provide a comprehensive analysis of the metabolomic alterations associated with two potent inducers of EMT in RPE. Despite both inducing EMT, TNFα and TGFβ2 appear to induce divergent metabolic alterations in RPE, highlighting the complexity of the interplay between metabolic reprogramming and EMT. Their divergent metabolomic profiles may be responsible, at least in part, for their differential impact on inflammation: TNFα induces a robust pro-inflammatory response in RPE [8], whereas TGFβ2 exhibits an anti-inflammatory effect [32].

Inflammation is closely intertwined with metabolic reprogramming. In our study, TNFα upregulated arachidonic acid, an ω-6 polyunsaturated fatty acid, which serves as a precursor to a cascade of pro-inflammatory eicosanoids, including prostaglandins and thromboxanes catalyzed by cyclooxygenase-1 and -2 enzymes, and leukotrienes catalyzed by the 5-lipoxygenase enzyme [33]. Arachidonic acid has been found to reduce the phagocytic capacity of RPE [34], and inhibition of 5-lipoxygenase protects RPE from sodium-iodate-induced degeneration [35]. TNFα can directly stimulate arachidonic acid in neutrophils, perpetuating the inflammatory cascade [36].

Intriguingly, we found that TNFα also significantly increased levels of the anti-inflammatory and antioxidant metabolite, nicotinamide (vitamin B3) [37]. In a metabolomics analysis of plasma samples from AMD patients vs. controls, nicotinamide was found to be significantly altered [38]. Nicotinamide potently suppresses complement activation and inflammation in RPE [39] and enhances mitochondrial metabolism to promote RPE differentiation [40]. A complex relationship exists between nicotinamide and TNFα. Paradoxical increases in levels of nicotinamide adenine dinucleotide (NAD^+^) have been associated with LPS-induced TNFα release in proinflammatory macrophages [41,42] and may be linked to dependence on the NAD^+^ salvage pathway to counteract the increased production of mitochondrial reactive oxygen species (ROS) [43]. It is possible that the increased nicotinamide seen with TNFα in RPE is a compensatory response that prolongs survival in the face of excessive TNFα-induced inflammation and oxidative stress.

Upregulation of the pentose phosphate pathway (PPP) by TNFα in RPE further supports the notion that antioxidants are upregulated to detoxify the increased ROS associated with TNFα. Diverting metabolic substrates from glycolysis into the PPP enables the generation of NADPH for antioxidant defense that serves as a cofactor required by glutathione reductase to reduce oxidized glutathione [44]. Moreover, the PPP supports reductive biosynthesis and ribose biogenesis [45], which are required for the cellular transformation involved in EMT. We also found secondary upregulation of the HIF-1 pathway by TNFα, which is a positive transcriptional regulator of PPP enzymes [46].

Our data showed that TGFβ2 significantly reduces malic acid, a key intermediate metabolite of the TCA cycle, corroborating our previous finding that TGFβ2 suppresses mitochondrial OXPHOS capacity in RPE [14]. In the TCA cycle, the conversion of malate to oxaloacetate is catalyzed by malate dehydrogenase (MDH) using NAD^+^ or NADP^+^ as a cofactor [47]. MDH1 is frequently overexpressed in cancer cells, where it has been shown to enhance glycolysis by replenishing the cytosolic cofactor NAD^+^ [48]. We previously found that exposure to TGFβ2 leads to similar metabolic rewiring with H-RPE showing enhanced glycolysis [14]. In mammals, there is a malate-succinate shuttle between the RPE and neural retina, in which malate exported from the RPE is imported into the retina to fuel succinate production [49,50]. Our data show that suppression of malate by TGFβ2 in RPE in vitro may have significant consequences for neighboring cells in vivo through disruption of the retinal metabolic ecosystem [51].

RPE phagocytose and digest the shed photoreceptor outer segment membranes that are rich in fatty acids [52]. Unsurprisingly, RPE exploit fatty acids as a key energy substrate and are highly dependent on fatty acid metabolism for ATP generation [53]. In this study, TGFβ2 treatment led to decreased levels of both palmitoleic acid and linoleic acid, indicating a disruption in fatty acid metabolism. Dysregulated lipid metabolism has been similarly linked to EMT in cancer; metastatic cells display increased lipolysis, which releases endogenous free fatty acids to generate the reduced form of NADPH through fatty acid oxidation (FAO) [54]. FAO confers a survival advantage for tumor cells by maintaining sufficient energy generation and redox homeostasis, as well as providing a source of lipids for membrane biogenesis [55]. It is possible that TGFβ2-treated RPE overutilize and deplete fatty acid supplies to enable increased FAO.

In conclusion, our results demonstrate the metabolic flexibility of RPE to rapidly adapt and rewire metabolic pathways during the induction of EMT. Despite both inducing EMT, TNFα and TGFβ2 trigger divergent metabolic signatures. While TNFα disrupted metabolic pathways that are involved in inflammation and oxidative stress, including arachidonic acid metabolism and the pentose phosphate pathway, TGFβ2 disrupted the TCA cycle and fatty acid oxidation. With a more complete understanding of the metabolites altered during retinal EMT, we can better comprehend the pathobiology of retinal pathologies such as AMD, PVR, and DR, as well as identify novel biomarkers for diagnosis, prognosis, treatment monitoring, and tailored drug development.

Future investigations will be aimed at exploring the endo- and exo-metabolomic profiles observed in pre-clinical in vivo models of AMD, PVR, and DR. A comparison of the differential metabolite changes from our study on TNFα and TGFβ2 in RPE with data obtained from in vivo models will better inform the individual contributions of these two EMT inducers. Furthermore, studies exploring changes in the metabolome of other retinal cell types involved in AMD, PVR, and DR, including endothelial cells and microglial, following exposure to TNFα and TGFβ2 would be of interest to clarify the contributions of different cell types to metabolomic alterations found in vivo. Identification of key disease-associated metabolic pathways will inform the design of drug screenings to assess the efficacy of novel pharmacotherapies in combating retinal fibrosis.

## Figures and Tables

**Figure 1 metabolites-13-00213-f001:**
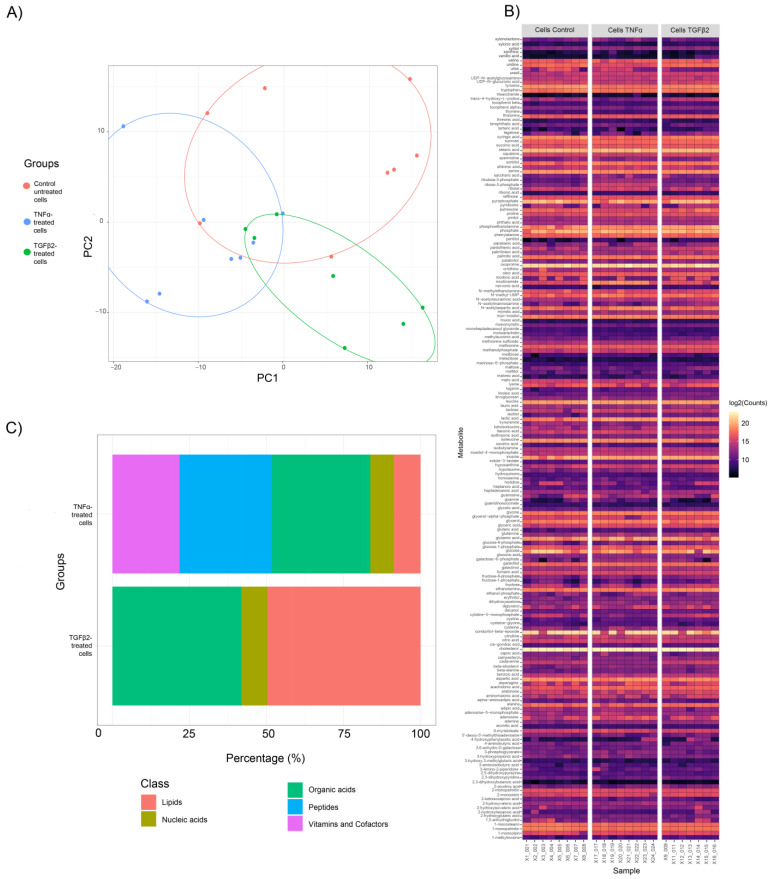
Metabolite profile clustering of TNFα- and TGFβ2-treated H-RPE cells compared to untreated control cells. (**A**) Principal Component Analysis (PCA) score plots show distinct clusters for samples in the control, TNFα and TGFβ2 cellular metabolite groups. (**B**) A heatmap of the differential fold changes for all annotated metabolites in each sample analyzed for the three groups. (**C**) Categorization of differentially expressed metabolites between TNFα- and TGFβ2-treated H-RPE cells compared to untreated control cells.

**Figure 2 metabolites-13-00213-f002:**
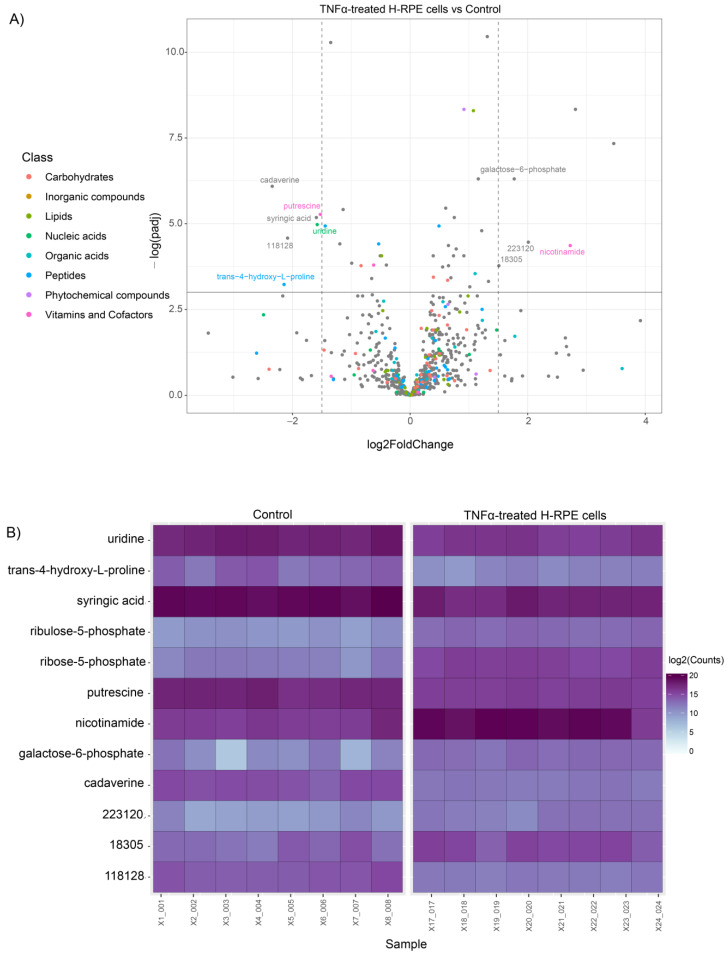
Differential metabolite analysis of TNFα-treated H-RPE cells vs. control. (**A**) A volcano plot illustrating the most significantly altered metabolites relative to fold change and statistical significance. (**B**) A heatmap of the most significantly altered metabolites for each sample analyzed for TNFα vs. control.

**Figure 3 metabolites-13-00213-f003:**
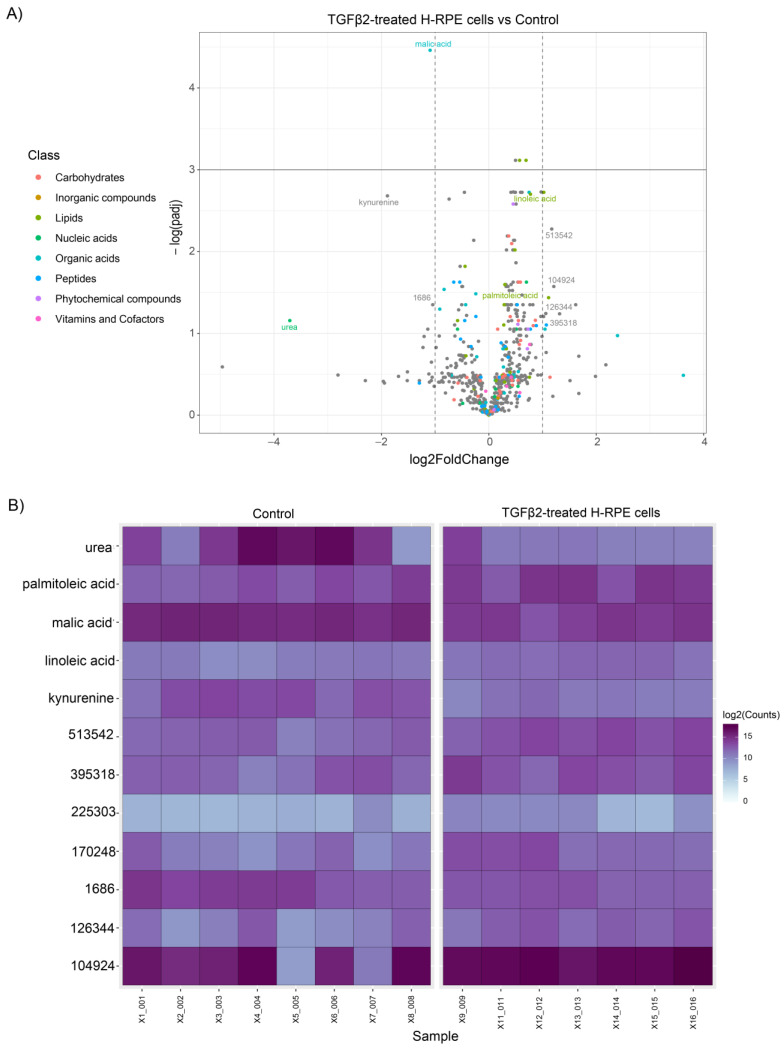
Differential metabolite analysis of TGFβ2-treated H-RPE cells vs. control. (**A**) A volcano plot illustrating the most significantly altered metabolites relative to fold change and statistical significance. (**B**) A heatmap of the most significantly altered metabolites for each sample analyzed for TGFβ2 vs. control.

**Figure 4 metabolites-13-00213-f004:**
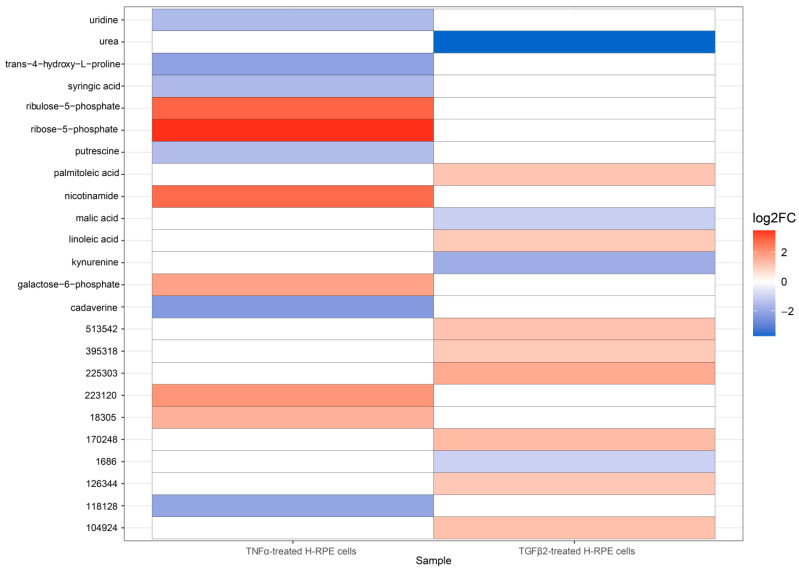
A heatmap depicting differential metabolite changes between TNFα-treated and TGFβ2-treated H-RPE cells. Upregulated metabolites are displayed in red, and downregulated metabolites are displayed in blue. Both annotated (primary) and non-annotated metabolites were included in this analysis. *p*adj < 0.05.

**Figure 5 metabolites-13-00213-f005:**
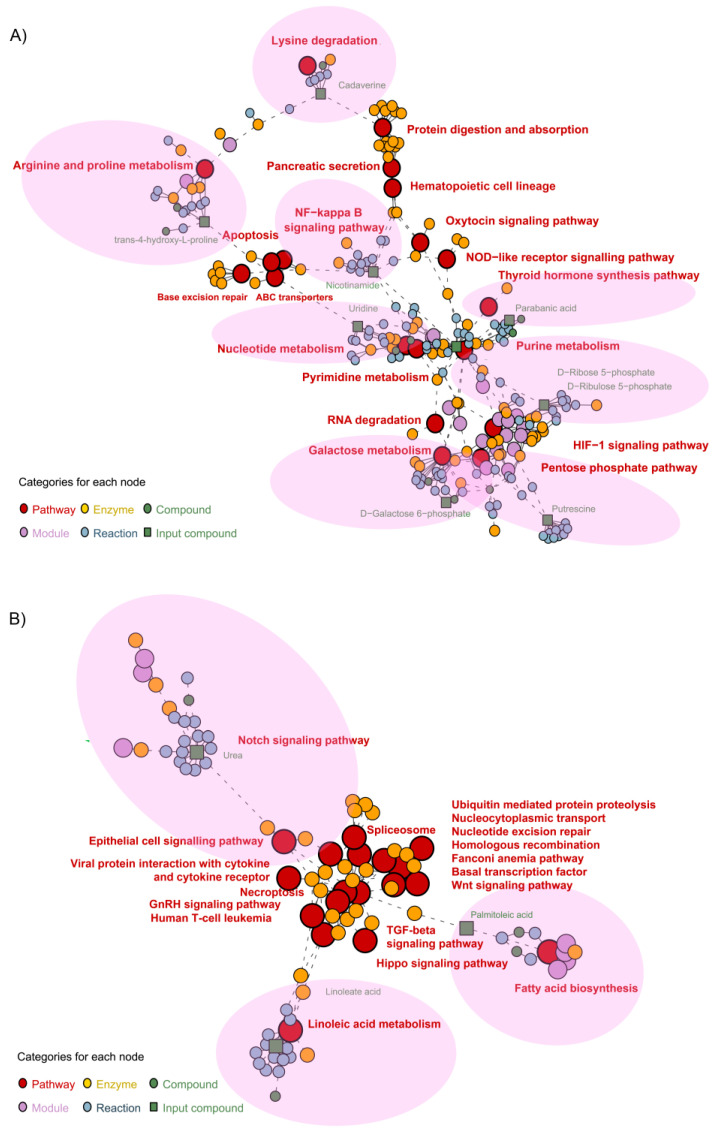
Metabolite pathway enrichment analysis of TNFα- and TGFβ2-treated H-RPE cells vs. control. Network enrichment analysis of the most differentially expressed metabolites for (**A**) TNFα- and (**B**) TGFβ2-treated H-RPE. Primary pathways affected are indicated by the pink shaded regions.

## Data Availability

All data will be made available upon request. The analysis code can be found on GitHub https://github.com/BioinfoHub-PeiQinNg/RetinalEMT_Metabolomics (accessed on 28 December 2022).

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
