# Peer review of "Divergent Metabolomic Signatures of TGFβ2 and TNFα in the Induction of Retinal Epithelial-Mesenchymal Transition"

_metabolites, 2023, doi:10.3390/metabo13020213_

Round 1

Reviewer 1 Report

In 'Divergent Metabolomic Signatures of TGFβ2 and TNFα in the 2 Induction of Retinal Epithelial-Mesenchymal Transition' the authors investigated the metabolomic profile of RPE cells undergoing EMT following TGFβ2 or TNFα treatment.

This is a well written manuscript with interesting results. The only comment I have is that l227 on page 5 should be Figure 5A and not 3A. Also, it might be helpful to add a Figure in the Suppl. files showing the EMT of the hRPE cells following the treatment.

Author Response

We thank the reviewer for the positive review and helpful suggestions. We apologize for the error and have now amended the text to state “Figure 5A”. We have previously published images of the morphological changes associated with EMT of H-RPE and have added text to direct readers to these references in the methods section in line 95.

Reviewer 2 Report

Summary:

In this excellent innovative and comprehensive metabolomic study, some of the different associated pathways are characterized that underlie TNFα and TGFβ2-induced epithelial-mesenchymal transition (EMT) in retinal pigment epithelial cells (RPE). Even though the upregulation of these cytokines contributes to inducing a host of different sight-compromising diseases, TNFα induces a robust pro-inflammatory response in RPE whereas TGFβ2 exhibits an anti-inflammatory effect. This difference is relevant because novel metabolic targets are sought to treat retinal diseases and selectively reduce inflammation. Their results show that these cytokines induce this EMT through divergent metabolic pathways. TNFα disrupts metabolic pathways inducing inflammation and oxidative stress that include arachidonic acid metabolism and the pentose phosphate pathway, whereas TGFβ2 disrupts the TCA cycle and fatty acid oxidation. Ultimately, these findings will abet efforts to tailor drug development selectively targeting novel biomarkers which can improve diagnosis, prognosis, and treatment monitoring of retinal diseases such as AMD, PVR, and DR.   There are no concerns regarding the experimental approaches and data interpretation of this groundbreaking study.

Author Response

We thank the reviewer for the positive review and kind words.

Reviewer 3 Report

The muscript is well written and well presented. However, the weakest points in the manuscripts are the introsuction and lack of conclusion section.

The introductions should avoids the fundemental basic science  and should outline novelty aspects and improtance of this study. More clear and specificroles of TNFα- and TGFβ2 in the pathogenesis of retinal diseases must be outlined. 

Conclusion section is missing and how this research can guide future research lines and pharmacological treatment of retinal diseases must be clearly identified. 

Author Response

We thank the reviewer for their kind words and valuable insights in enhancing our manuscript. We have now expanded the introduction to emphasize the novel aspects of this study and clarified the specific roles of TNFα and TGFβ2 in retinal fibrosis. We have also added more discussion of future research directions and pharmacotherapies for the field of retinal fibrosis in the concluding paragraph.

Round 2

Reviewer 3 Report

The authors have satisfactorily addressed the comments.